# The Modulation of *SCO2730/31* Copper Chaperone/Transporter Orthologue Expression Enhances Secondary Metabolism in Streptomycetes

**DOI:** 10.3390/ijms221810143

**Published:** 2021-09-20

**Authors:** Nathaly González-Quiñónez, Ignacio Gutiérrez-Del-Río, Paula García-Cancela, Gemma Fernández-García, Sergio Alonso-Fernández, Paula Yagüe, Álvaro Pérez-Valero, María Montes-Bayón, Felipe Lombó, Ángel Manteca

**Affiliations:** 1Área de Microbiología, Departamento de Biología Funcional, IUOPA, ISPA, Facultad de Medicina, Universidad de Oviedo, 33006 Oviedo, Spain; natygq@gmail.com (N.G.-Q.); nachogutiem@gmail.com (I.G.-D.-R.); gemmafg06@hotmail.com (G.F.-G.); sergioalonsofernandez@gmail.com (S.A.-F.); yaguepaula@uniovi.es (P.Y.); apv.moratalla@gmail.com (Á.P.-V.); lombofelipe@uniovi.es (F.L.); 2Department of Physical and Analytical Chemistry, ISPA, Faculty of Chemistry, Universidad de Oviedo, 33006 Oviedo, Spain; garciacpaula@uniovi.es (P.G.-C.); montesmaria@uniovi.es (M.M.-B.)

**Keywords:** *Streptomyces*, differentiation, copper, secondary metabolism

## Abstract

Streptomycetes are important biotechnological bacteria that produce several clinically bioactive compounds. They have a complex development, including hyphae differentiation and sporulation. Cytosolic copper is a well-known modulator of differentiation and secondary metabolism. The interruption of the *Streptomyces coelicolor SCO2730* (copper chaperone, *SCO2730::Tn5062* mutant) blocks *SCO2730* and reduces *SCO2731* (P-type ATPase copper export) expressions, decreasing copper export and increasing cytosolic copper. This mutation triggers the expression of 13 secondary metabolite clusters, including cryptic pathways, during the whole developmental cycle, skipping the vegetative, non-productive stage. As a proof of concept, here, we tested whether the knockdown of the *SCO2730/31* orthologue expression can enhance secondary metabolism in streptomycetes. We created a *SCO2730/31* consensus antisense mRNA from the sequences of seven key streptomycetes, which helped to increase the cytosolic copper in *S. coelicolor*, albeit to a lower level than in the *SCO2730::Tn5062* mutant. This antisense mRNA affected the production of at least six secondary metabolites (CDA, 2-methylisoborneol, undecylprodigiosin, tetrahydroxynaphtalene, α-actinorhodin, ε-actinorhodin) in the *S. coelicolor*, and five (phenanthroviridin, alkylresorcinol, chloramphenicol, pikromycin, jadomycin G) in the *S. venezuelae*; it also helped to alter the *S. albus* metabolome. The *SCO2730/31* consensus antisense mRNA designed here constitutes a tool for the knockdown of *SCO2730/31* expression and for the enhancement of *Streptomyces*’ secondary metabolism.

## 1. Introduction

Streptomycetes are important biotechnological bacteria, producing two thirds of the bioactive secondary metabolites currently used in clinical applications (mainly antibiotics, but also antitumourals, immunosuppressors, etc.) [1,2]. They have complex developmental cycles that include programmed cell death (PCD), hyphae differentiation and sporulation [3,4].

Secondary metabolism in streptomycetes is highly regulated. There are regulators of specific secondary metabolism pathways as well as pleiotropic regulators affecting the production of several secondary metabolites [5]. There are also numerous predicted secondary metabolism pathways that have not been activated in the laboratory (cryptic pathways) [6,7]. Secondary metabolism is tightly connected with hypha differentiation and sporulation [8,9], which makes *Streptomyces*’ secondary metabolism even more complex. Understanding secondary metabolism regulation is one of the main challenges in industrial microbiology, and one of our best chances to activate the expression of cryptic pathways in the laboratory [10].

Copper is a well-known modulator of *Streptomyces* differentiation (aerial mycelium and sporulation) and secondary metabolism (antibiotic production) [11,12,13,14,15]. Cytosolic copper is highly regulated by copper chaperones and transporters, whose genetic expression is tightly modulated [16]. In a recent study, we quantified cytosolic copper in hypha and single spores, revealing that cytosolic copper correlates with spore germination, hypha differentiation and secondary metabolism activation [13,17]. The interruption of the *S. coelicolor SCO2730* copper chaperone ORF (*SCO2730::Tn5062* mutant) blocks *SCO2730* and reduces the expression of *SCO2731* (P-type ATPase copper export transporter), increasing cytosolic copper during germination and in the substrate mycelium hyphae (vegetative mycelium) [13]. There is a correlation between cytosolic copper, secondary metabolism activation and sporulation: low cytosolic copper concentrations are present in the vegetative hyphae; slightly higher cytosolic copper concentrations are present in secondary metabolite producer hyphae; and the highest cytosolic copper levels are present in sporulating hyphae and dormant spores [13]. Cytosolic copper in the *S. coelicolor SCO2730::Tn5062* mutant is always above the necessary threshold for the activation of secondary metabolism, generating a unique phenotype consisting of the production of secondary metabolites (undecylprodigiosin for instance) during its whole developmental cycle, including germination and substrate mycelium; in other words, this mutant lacks an authentic vegetative non-producing stage [13]. The *SCO2730::Tn5062* mutant shows a dramatic enhancement in the expression of genes from 13 secondary metabolite clusters (43.3% of all predicted secondary metabolites produced by this strain), including six genetic clusters that were predicted to participate in secondary metabolite biosynthesis, yet were never observed under the usual laboratory cultivation conditions (cryptic pathways) [13].

The aim of this work was to reproduce the phenotype of the *S. coelicolor SCO2730::Tn5062* copper chaperone/transport mutant (activation of secondary metabolism [13]) in other streptomycetes. The *SCO2730/31* genes are highly conserved in *Streptomyces* (80.6 and 80.7% average nucleotide identity, respectively, among *S. coelicolor*, *S. griseus*, *S. avermitillis*, *S. lividans*, *Saccharopolyspora erythraea* (formerly *Streptomyces erythraeus*), *S. tsukubaensis* and *S. venezuelae*) [13]. However, the *SCO2730/31*’s surrounding regions are not conserved enough to design a mutagenesis protocol capable of knocking out the *SCO2730/31* orthologues in any streptomycete. The construction of an *SCO2730::Tn5062* orthologue mutant for each streptomycete would be difficult to make scalable in high-throughput screening (HTS) campaigns. In this work, we opted for the creation of an antisense mRNA against a consensus sequence obtained from the *SCO2730/31* orthologues from the seven streptomycetes indicated above. We tested this antisense mRNA in *S. coelicolor*, the *Streptomyces* model strain characterised by the production of coloured actinorhodin (blue) and undecylprodigiosin (red) antibiotics [18]; *S. venezuelae*, the strain from which chloramphenicol was discovered [19]; and *S. albus*, a strain widely used as a heterologous host [20]. The antisense *SCO2730/31* consensus mRNA created here was successful in the modification of secondary metabolism in both *Streptomyces* species and constitutes a tool for knockdown *SCO2730/31* orthologue expression in order to enhance secondary metabolism in streptomycetes.

## 2. Results

### 2.1. Construction of the S. coelicolor, S. venezuelae and S. albus SCO2730/31 Orthologue Knockdown Strains

We designed an antisense mRNA against a consensus sequence of the *S. griseus*, *S. avermitillis*, *S. lividans*, *S. clavuligerus*, *Saccharopolyspora erythraea*, *S. tsukubaensis* and *S. venezuelae SCO2730/31* orthologues consisting of 2518 bp (Figure 1a–d, Appendix A). This sequence included the *SCO2730* and *SCO2731* ORF orthologues (Figure 1a,b) as well as two consensus sequences upstream of the *SCO2730/31* orthologue ORFs, long enough to include the gene ribosomal binding sites (Figure 1c,d). The homology of the *SCO2730/31* orthologue ORF sequences was over 50% for most of the consensus sequence positions (Figure 1a,b), which was considerably higher than the homology of the RBS sequences (Figure 1c,d). As a control, we also designed the *S. coelicolor SCO2730/31* sequence that was 100% homologous to the *S. coelicolor SCO2730/31* and their respective RBSs (2590 bps, Appendix A).

As detailed in the methods section, the *SCO2730/31* antisense mRNA minus chain sequences were cloned into the ΦBT1 integrative pNG4-SP44 plasmid under the control of the SP44 strong constitutive promoter [21] (Figure 1e). Both constructions were used to knock down the *SCO2730/31* gene expression in *Streptomyces coelicolor*. The pNG4-SP44 expressing the *SCO2730/31* consensus antisense mRNA was also used to knock down the expression of the *SCO2730/31* orthologues in *S. venezuelae* and *S. albus*.

### 2.2. S. coelicolor SCO2730/31 Knockdown Mutants Affected Germination, Secondary Metabolite Production and Cytosolic Copper Levels

The original *SCO2730::Tn5062* copper chaperone/transport mutant, whose phenotype we wished to reproduce, showed as its main characteristics a delayed germination, a massive secondary metabolism activation and a high cytosolic copper accumulation [13]. To analyse the phenotypical effects of the antisense mRNAs, we compared the phenotype of the two *S. coelicolor SCO2730/31* knockdown strains (those expressing the antisense mRNAs) against the *SCO2730::Tn5062* mutant [13] and the *S. coelicolor* wild-type strain with and without the pNG4-SP44 plasmid (Figure 2).

Antibiotic production and germination were significantly affected in the *SCO2730/31* knockdown mutants (Figure 2a,b). However, the effects in these phenotypes were different in the *SCO273031* knockdown mutants and the original *SCO2730::Tn5062* mutant. The knockdown mutants demonstrated a lower increase in antibiotic production (compared to the wild-type strain with and without the pNG4-SP44 plasmid) than the *SCO2730::Tn5062* mutant (Figure 2a). The effect on antibiotic production was lesser in the *SCO2730/31* knockdown mutant expressing the *S. coelicolor* antisense mRNA (100% homologue to *S. coelicolor*) than in the *SCO2730/31* knockdown mutant expressing the consensus mRNA; there was no significant increase in actinorhodin production in the latter mutant (Figure 2a). Germination was reduced in both *SCO2730/31* knockdown mutants compared to the wild-type strain with and without pNG4-SP44, but the reduction was lower than that in the *SCO2730::Tn5062* mutant (Figure 2b).

As reported previously [13], cytosolic copper in dormant spores was significantly higher in the *SCO2730::Tn5062* mutant compared to the wild-type strain (Figure 2c). However, there were no significant differences between the cytosolic copper in the spores of the knockdown mutants and in the wild-type strain (Figure 2c). Cytosolic copper in young hyphae was also higher in the *SCO2730::Tn5062* mutant compared to the wild-type strain. Surprisingly, cytosolic copper in young hyphae was slightly increased in the wild-type strain harbouring the empty pNG4-SP44 plasmid (Figure 2c, middle panel), albeit to a considerably lower extent than in the *SCO2730/31* knockdown mutant expressing the consensus mRNA (Figure 2c, right panel). The increase in cytosolic copper in the young hyphae of the *SCO2730/31* knockdown mutant expressing the *S. coelicolor* antisense mRNA (100% homologue to *S. coelicolor*) was not significant compared to the wild-type strain harbouring the pNG4-SP44 plasmid (Figure 2c, right panel).

### 2.3. SCO2730/31 Expression Knockdown had a Large Effect on the S. coelicolor Metabolome, Albeit to a Lesser Extent than the Effect Observed in the SCO2730::Tn5062 Mutant

As the increase in actinorhodin and undecylprodigiosin production of the *SCO2730/31* antisense consensus mRNA were higher than the effect of the *S. coelicolor* SCO2730/31 antisense mRNA (Figure 2a), the detailed metabolome analyses were performed only in the *S. coelicolor* strain overexpressing the *SCO2730/31* antisense consensus mRNA. We compared the metabolomes of the *S. coelicolor* wild-type strain against the *S. coelicolor* harbouring the empty pNG4-SP44 plasmid (pNG4-SP44 strain), the *S. coelicolor* harbouring pNG4-SP44 expressing the consensus *SCO2730/31* antisense RNA (*SCO2730/31* knockdown strain) and the original *SCO2730::Tn5062* mutant [13]. The results of the comparison of the metabolomes of three biological replicates, obtained at 145 h from sucrose-free liquid R5A cultures (the time-point at which maximum antibiotic production was reached), are summarised in Figure 3. In total, 432 compounds showed significant differences in the *S. coelicolor* strain harbouring the empty pNG44-SP44 plasmid compared to the wild-type strain (q-value less than 0.05 and fold change over 2.5-fold); 1699 compounds showed significant differences between the wild-type strain and the *S. coelicolor SCO2730/31* knockdown; and 7448 compounds showed significant differences between the wild-type strain and the *SCO2730::Tn5062* mutant (Figure 3a, Appendix A).

Among the compounds showing significant differences between the *S. coelicolor SCO2730/31* mutants analysed and the wild-type strain, we identified, based on the bibliography [22] and their exact masses (±0.001 Da maximum allowed variation), putative LC-MS *m*/*z* ions from at least 14 secondary metabolites (Figure 3b): actinorhodin, coelimycin, flaviolin, γ-butyrolactone, germicidin A, germicidin B, germicidin D, calcium-dependent antibiotic (CDA), 2-methylisoborneol, undecylprodigiosin, tetrahydroxynaphtalene, α-actinorhodin, γ-actinorhodin and ε-actinorhodin (Figure 3b). The production of all these 14 compounds was affected in the *SCO2730::Tn5062* mutant compared to the wild-type strain. CDA, 2-methylisoborneol, undecylprodigiosin and tetrahydroxynaphtalene production was altered in the *S. coelicolor SCO2730/31* knockdown mutant; α- and ε-actinorhodins were affected in the *SCO2730::Tn5062* and *SCO2730/31* knockdown mutants, as well as in the wild-type strain harbouring the pNG4-SP44 empty plasmid. Among the other eight compounds, secondary metabolite production was only affected in the *SCO2730::Tn5062* mutant (Figure 3b).

A detailed analysis of the abundance of the specific isomers and adducts of the secondary metabolites putatively identified, showing significant differences between the *S. coelicolor SCO2730/31* mutants and the wild-type strain, is shown in Figure 3c. The abundance of two compounds was affected in *S. coelicolor* [pNG4-SP44], 10 in the *S. coelicolor SCO2730/31* knockdown mutant and 88 in the *SCO2730::Tn5062* mutant (Appendix A). Interestingly, the differences in secondary metabolism between the two *SCO2730/31* mutants analysed and the wild-type strain were greater than those that can be inferred from Figure 3b. For instance, α-actinorhodin showed significant differences in all strains compared to the wild-type strain (Figure 3b), but there was a specific adduct, α actinorhodin (M+Na-2H) (1-) (compound number **6** in Figure 3c), that was significantly overproduced in the *SCO2730/31* knockdown strain and not in the *SCO2730::Tn5062* mutant or the wild-type strain harbouring pNG4-SP44 (Figure 3c).

The chromatograms of some representative secondary metabolites are shown in Figure 4. Most of the identified secondary metabolites were significantly up-regulated in the *SCO2730::Tn5062* mutant compared to the other strains: coelimycin (compound **11**), CDA (compound **12**), two isomers of tetrahydronaphtalene (compounds **13** and **15**), actinorhodin (compound **16**), α-actinorhodin (2M+HAc-H) (1-) (compound **10**), γ-butyrolactone (compounds **17** and **18**), flaviolin (M+HAc-H)- (compound **14**), ε-actinorhodin (compound **16**) and γ-actinorhodin (compound **19**) (Figure 4a,d). The abundance of one of the tetrahydronaphtalene isomers detected (compound **13**), was significantly up-regulated in the *SCO2730::Tn5062* and the *SCO2730/31* knockdown mutants (Figure 4a). Interestingly, the abundance of coelimycin (compound **11**) and CDA (compound **12**), that were significantly up-regulated in the *SCO2730::Tn5062* mutant (Figure 4a) seemed to be up-regulated in the *SCO2730/31* knockdown mutant as well, but without sufficient reproducibility to be statistically significant. By contrast, some compounds were significantly down-regulated in the *SCO2730::Tn5062* mutant (germicidins A, B and D, and compounds **7**–**9**; Figure 4f). Other compounds were overproduced in the *S. coelicolor SCO2730/31* knockdown mutant, such as 2-methylisobormeol (M+HAc-H) (1-) (compound **5**), α-actinorhodin (M+Na-2H) (1-) (compound **6**) and undecylprodigiosin (compound **3**) (Figure 4e). This last result coincided with the abundance of the undecylprodigiosin production quantified spectrophotometrically, which was slightly more abundant in the *S. coelicolor SCO2730/31* knockdown mutant (expressing the consensus antisense mRNA) than in the *SCO2730::Tn5062* mutant (Figure 2a). The production of a unique compound, ε-actinorhodin (compound **1**), was significantly up-regulated in the *S. coelicolor* strain harbouring pNG4-SP44 (Figure 4c). The only compound identified as significantly down-regulated in all the analysed strains compared to the wild-type strain was α-actinorhodin (compound **2**; Figure 4b).

### 2.4. The SCO2730/31 Expression Knockdown had a Big Effect on the S. venezuelae Metabolome

Subsequently, we analysed the effect of the *SCO2730/31* expression knockdown in *S. venezuelae*. We compared the metabolomes of the *S. venezuelae* wild-type strain against the *S. venezuelae* harbouring the empty pNG4-SP44 plasmid (pNG4-SP44 strain) and the *S. venezuelae* harbouring the pNG4-SP44 plasmid expressing the consensus *SCO2730/31* antisense RNA (*SCO2730/31* knockdown strain). The results of the metabolomes of three biological replicates of these strains, obtained at 48 h (the time point at which maximum antibiotic production was reached) from liquid MYM medium cultures [23] (supplemented with 2.1 g/L MOPS), are shown in Figure 5. In total, 6438 compounds showed significant differences (q-value less than 0.05 and fold change greater than 2.5-fold) in the *S. venezuelae* strain harbouring the empty pNG4-SP44 plasmid compared to the wild-type strain, whereas 4792 compounds showed significant differences between the *S. venezuelae SCO3730/31* knockdown and the wild-type strain. As discussed below, this result reveals a strong effect of the pNG4-SP44 plasmid on the *S. venezuelae* metabolome.

Among the compounds showing significant abundance differences in the *S. venezuelae* pNG4-SP44 and the *S. venezuelae SCO2730/31* knockdown strains compared to the wild-type strain, we identified the LC-MS *m*/*z* ions of six putative secondary metabolites: ectoine varied in the pNG4-SP44 strain; pikromycin and jadomycin G varied in both the pNG4-SP44 and the *SCO2730/31* knockdown strains; and phenanthroviridin, alkylresorcinol and chloramphenicol varied in the *S. venezuelae SCO2730/31* knockdown strain (Figure 5b).

A detailed analysis of the abundance of the specific HPLC-MS secondary metabolite *m*/*z* ions differentially produced in the pNG4-SP44 and the *SCO2730/31* knockdown strains compared to the wild-type strain (Figure 5c; Appendix A) revealed that the differences were higher than those that can be inferred from Figure 5b. Only one secondary metabolite was identified as differentially produced in both the pNG4-SP44 and the *SCO2730/31* knockdown strains (pikromycin [M+TFA-H] (1-), compound **2**, labelled in red in Figure 5c). The other 15 differential secondary metabolite ions were not shared between the pNG4-SP44 (eight compounds) and the *SCO2730/31* knockdown mutant (seven compounds) (Figure 5c).

Figure 6 shows the chromatograms of some of the HPLC-MS secondary metabolite *m*/*z* ions differentially produced in the pNG4-SP44 and the *SCO2730/31* knockdown strains compared to the wild-type strain. Ectoine [M+FA-H] (1-) was produced in higher amounts in the pNG4-SP44 strain (compound **1** in Figure 6a), whereas phenanthroviridin [M+HAc-H] (1-) and alkylresorcinol were more abundant in the *SCO2730/31* knockdown (compounds **6** and **7** in Figure 6c). Chloramphenicol production was also affected; in this case, it was reduced in the *SCO2730/31* knockdown mutant but not in the strain harbouring the empty pNG4-SP44 plasmid (compound **5** in Figure 6d). The Jadomycin G chromatogram revealed that the same mass was identified at different retention times in the pNG4-SP44 (compounds **3** in Figure 6e) and the *SCO2730/31* knockdown (compound **4** in Figure 6e) strains, indicating the presence of different isomers (Figure 6e). Pikromycin (M+TFA-H) (1-) (compound **2**) was the only compound whose production was reduced in both the *S. venezuelae* strains harbouring the pNG4-SP44 plasmid with and without the consensus antisense *SCO2730/31* mRNA.

### 2.5. The SCO2730/31 Expression Knockdown Modified the S. albus Metabolome

Finally, we analysed the effect of the *SCO2730/31* expression knockdown in *S. albus*. We compared the metabolomes of the *S. albus* wild-type strain against the *S. albus* harbouring the empty pNG4-SP44 plasmid (pNG4-SP44 strain) and the *S. albus* harbouring the pNG4-SP44 plasmid expressing the consensus *SCO2730/31* antisense RNA (*SCO2730/31* knockdown strain). In order to perform a quick analysis, the metabolomes were compared using high-performance liquid chromatography coupled with diode-array detection (HPLC-DAD) (Figure 7). The HPLC-DAD chromatograms were almost identical in the *S. albus* wild-type strain and the *S. albus* harbouring the empty pNG4-SP44 plasmid (Figure 7). However, there were several differences between the HPLC-DAD chromatograms of the wild-type and the *S. albus SCO2730/31* knockdown strains (rectangles in Figure 7a, magnifications in Figure 7b,c).

## 3. Discussion

As introduced above, the *S. coelicolor SCO2730::Tn5062* mutant blocks the expression of the *SCO2730* copper chaperone and reduces the expression of the *SCO2731* copper transporter, augmenting cytosolic copper and increasing the transcription of 13 secondary metabolite clusters (43.3% of all predicted secondary metabolites produced by this strain) [13]. These secondary metabolites include six genetic clusters that were predicted to participate in secondary metabolite biosynthesis, yet were never observed under laboratory cultivation conditions (cryptic pathways) [13]. In this work, we performed a metabolomic analysis of this mutant, confirming that the activation of the transcription of secondary metabolite genes correlates with a massive increase in secondary metabolite production. The production of 88 putatively identified secondary metabolites was affected compared to the wild-type strain. Seventy-six of these compounds (86% of the total), were significantly more abundant in the mutant than in the wild-type strain (Figure 3c; Appendix A).

The regulation of the *SCO2730/31*’s gene expression is highly complex and modulated by at least four promoters [13]. The *SCO2730::Tn5062* mutant is interrupted at the beginning of the *SCO2730* ORF, allowing a reduced expression of *SCO2731* from a promoter located into *SCO2102* [13]. Further unknown expression regulation networks cannot be discarded, since different combinations of the four promoters, together with the *SCO2102/03* ORFs, cannot fully restore the *SCO2730/31* transcription in the *SCO2730::Tn5062* mutant [13]. The objective of this work was to reproduce the phenotype (secondary metabolism and cryptic pathway activation) of the *SCO2730::Tn5062* mutant in other streptomycetes. To create the mutant in the corresponding *SCO2730* orthologue for each streptomycete species would be tedious and unfeasible in high-throughput screening using several strains. Here, we designed a proof of concept consisting of a test of whether an antisense mRNA for a consensus sequence of the *SCO2730/31* orthologue genes can successfully activate *Streptomyces*’ secondary metabolism. The three *S. coelicolor* strains mutated in *SCO2730/31* (*SCO2730::Tn5062*; *SCO2730/31* knockdown expressing consensus antisense mRNA; *SCO2730/31* knockdown expressing the *S. coelicolor* antisense mRNA) were different, and would conceivably promote different SCO2730/31 copper chaperone/transporter doses. We could not perform a direct quantification of the SCO2730/31 protein amounts, but we quantified significant differences in cytosolic copper levels between the mutants and the wild-type strain (Figure 2c), which is an indirect indicator of the differences in the SCO2730/31 copper transporter activity. The differences in cytosolic copper lead to a different effect on actinorhodin and undecylprodigiosin production (Figure 2). These results support the copper dose-dependent effect described by us in a previous study: secondary metabolism is activated in hypha harbouring cytosolic copper levels between 46 and 200 ng Cu/mg protein, but it is inhibited at the high cytosolic copper levels reached during sporulation [13]. Theoretically, the antisense *SCO2730/31* mRNA 100% homologous to the *S. coelicolor SCO2130/31* genes should hybridise more tightly to the *SCO2730/31* mRNA than the consensus antisense mRNA, which is not identical to the *S. coelicolor* genes. However, the lower effect on cytosolic copper accumulation and antibiotic production of the former seems to indicate the contrary, i.e., a lower effect on the reduction of *SCO2730/31* translation in *S. coelicolor*.

The *S. coelicolor SCO2730/31* knockdown mutant overexpressing the consensus antisense mRNA significantly affected the production of 1699 compounds (Figure 3a), including 10 HPLC-MS *m*/*z* ions from six putatively identified secondary metabolites (Figure 3b,c). The antisense consensus mRNA also worked to modify the *Streptomyces venezuelae* metabolome (Figure 5). Although the empty pNG4-SP44 plasmid had a large effect on the *S. venezuelae* metabolome, significantly affecting the production of 6438 compounds (Figure 5a), there were important differences between the *S. venezulae* harbouring pNG4-SP44 and the *S. venezuelae* knockdown mutant metabolomes, revealing that the consensus antisense mRNA also worked to alter secondary metabolism in *S. venezuelae*: of the six secondary metabolites from which an isomer or adduct was putatively identified, three (phenanthroviridin, alkylresorcinol and chloramphenicol) significantly affected production only in the strain expressing the consensus antisense mRNA (Figure 5b); pikromycin (M+TFA-H) (1-) was the only adduct that was significantly altered in the *S. venezuelae* harbouring the pNG4-SP44 with and without the *SCO2730/31* consensus antisense mRNA. The production of the other 16 HPLC-MS *m*/*z* secondary metabolite ions putatively identified was different in the *S. venezuelae SCO2730/31* knockdown mutant and the *S. venezuelae* strain harbouring the empty pNG4-SP44 plasmid.

The *S. albus SCO2730/31* knockdown mutant overexpressing the consensus antisense mRNA showed an important alteration in its metabolome (Figure 7). Unlike the *S. venezuelae*, the empty pNG4-SP44 plasmid did not have a significant effect on the *S. albus* metabolome (Figure 7).

## 4. Materials and Methods

### 4.1. Bacterial Strains and Culture Conditions

All the *Streptomyces* and *Escherichia coli* strains used in this work are listed in Appendix A.

The *S. coelicolor* spores were harvested from SFM solid plates [18] after growth at 30 °C for 7 days. The *S. coelicolor* germination was analysed in GYM [24] (5 g/L glucose, 4 g/L yeast extract, 5 g/L malt extract, 0.5 g/L MgSO_4_·7H_2_O, 20 g/L agar; after autoclaving supplemented with sterile 0.5 g/L K_2_HPO_4_) plates covered with cellophane, inoculated with 10^7^ spores from a freshwater suspension and cultured at 30 m °C. The fermentation was performed in liquid 50 mL sucrose-free R5A [25] cultures grown at 30 °C and 200 rpm in 250 mL flasks inoculated with 10^7^ spores/mL.

The *S. venezuelae* spores were harvested from MYM solid plates [23] after growth at 30 °C for 8 days. The pre-inoculum for the fermentation was prepared using 5.10^6^ spores from a freshwater suspension in 10 mL of TSB medium (tryptic soy broth, Scharlau), and grown for 16 h at 30 °C and 220 rpm in 100 mL flasks. The fermentation was performed in 20 mL MYM medium supplemented with 2.1 g/L MOPS and inoculated with 2 mL of pre-inoculum, at 30 °C and 220 rpm in 250-mL flasks. After 8 h of incubation, absolute ethanol was added to each culture to obtain a concentration of 6% *v*/*v*.

The *S. albus* spores were harvested from Bennett medium solid plates [18] after growth at 30 °C for 5 days. The fermentation was performed in liquid 50 mL NL333 medium [13] cultures grown at 30 °C and 250 rpm in 250 mL flasks inoculated with 10^7^ spores/mL.

The *Escherichia coli* strains were cultured in LB and 2xTY media at 37 °C. The following antibiotics were added to select plasmid-bearing and mutant strains: ampicillin (100 μg/mL), apramycin (100 μg/mL for *E. coli*, 25 µg/mL for *S. coelicolor*), hygromycin (100 μg/mL for *E. coli*, 200 µg/mL for *S. coelicolor*), kanamycin (50 μg/mL), chloramphenicol (25 μg/mL) and nalidixic acid (25 µg/mL) (all from Invitrogen, Waltham, MA, USA).

### 4.2. SCO2730/31 Bioinformatic Analyses and Antisense mRNA Design

The *SCO02730/2731* orthologue sequences and their surrounding genomic regions were analysed. The *S. coelicolor*, *S. griseus*, *S. avermitillis*, *S. lividans* and *S. venezuelae* orthologue sequences were obtained from the StrepDB (http://strepdb.streptomyces.org.uk/) (accessed on 10 June 2021). The *Saccharopolyspora erythraea* NRRL 2338 and *S. tsukubaensis* NRRL18488 *SCO02730/2731* orthologue sequences were obtained from NCBI (genome accession numbers AM420293 and NZ_CP029159, respectively).

The SCO2730 orthologues were: SLI_3079 (*S. lividans*), SAV_5332 (*S. avermitilis*), SVEN_2533 (*S. venezuelae*), SGR_4828 (*S. griseus*), SACE_6610 (*S. erythraea*) and STSU_RS23720 (*S. tsukubaensis*).

The SCO2731 orthologues were: SLI_3080 (*S. lividans*), SAV_5331 (*S. avermitilis*), SVEN_2534 (*S. venezuelae*), SGR_4827 (*S. griseus*), SACE_6611 (*S. erythraea*) and STSU_RS20065 (*S. tsukubaensis*).

The sequences containing the putative *SCO2730* and *SCO2731* RBS sequences were obtained by collecting 20 nucleotides upstream of the ATG (or GTG) starting codon.

The *SCO2730* and *SCO2731* nucleotide similarities were estimated using the software package Lalign (http://www.ch.embnet.org/software/LALIGN_form.html) (accessed on 10 June 2021).

The *SCO2730*, *SCO2731* and RBSs’ DNA sequences were aligned using the MUSCLE software available on the free online platform Phylemon (http://phylemon.bioinfo.cipf.es/) (accessed on 10 June 2021). Ambiguous alignments in highly variable (gap-rich) regions were excluded from the antisense mRNA sequence (aligned sequences are available from the authors upon request).

The consensus antisense mRNA sequence (2518 bps) was generated by fusing the consensus sequences of the four alignments (SCO2730, SCO2731 and the two RBS regions) (Appendix A). The homology plots shown in Figure 1a–c were created using the Jalview 2.11.0 software. A sequence 100% homologous to the *S. coelicolor SCO2730/31* and RBS sequences (2590 bps) was also designed (Appendix A). The synthesis of both sequences was ordered by BGI Genomics (Hong Kong).

### 4.3. Overproduction of the SCO2730/31 Antisense mRNA

The *SCO2730/31* consensus and *S. coelicolor* antisense (100% homologous to *S. coelicolor*) mRNAs were cloned, independently, in the pNG4 plasmid at the *Spe*I–*Nde*I cloning sites [13]. To increase the antisense mRNA production, the strong SP44 promoter [21] was cloned into this plasmid to control the overproduction of the antisense mRNA. In order to do this, we ordered the synthesis of the *Nde*I- *fd-ter* terminator-SP44-RBS SR41-*Nde*I DNA from BGI Genomics (Hong Kong). The *fd-ter* terminator [21] was included before the SP44 promoter to prevent its possible interference with the *PermE* * promoter present in the original pNG4 plasmid. The SR41 RBS [21] is not necessary for the expression of the antisense mRNA, but it was included in order to obtain the possibility of using this plasmid to overproduce proteins. The synthetised sequence was cloned into *Nde*I, and the correct orientation of the SP44 promoter, controlling the expression of the antisense mRNAs, was checked by *Xho*I digestion and confirmed by Sanger sequencing, using SP44F/R primers. The integration of the plasmid in the *S. coelicolor* and *S. venezulae* chromosomes was confirmed by PCR using the primers SCO4848F/pMS82R [26] and SCO4848F/vnz22340, respectively.

### 4.4. Undecylprodigiosin and Actinorhodin Quantification

Undecylprodigiosin and actinorhodin were quantified spectrophotometrically, in accordance with Tsao et al. [27] and Bystrykh et al. [28]. For actinorhodin quantification, KOH was added to the culture samples at a final concentration of 1N. The cellular pellets were discarded by centrifugation, and the actinorhodin concentration was spectrophotometrically determined at 640 nm, applying the linear Beer-Lambert relationship (*ε*_640_ = 25,320). The culture samples for undecylprodigiosin quantification were lyophilised, resuspended in methanol, acidified with 0.5N HCl and spectrophotometrically assayed at 530 nm, using the Beer–Lambert relationship to estimate concentrations (*ε*_530_ = 100,500). Three biological replicates were processed. The reliability of the differences in antibiotic production (compared to the wild-type strain) was analysed by Student’s *t*-tests. Differences were considered significant if the *p* value was equal to or less than 0.05 (asterisks in Figure 2a).

### 4.5. Protein Quantification

The protein concentration was quantified by Bradford [29], using bovine serum albumin standard (Sigma Aldrich, Burlington, MA, USA). The total protein extracts were obtained by mixing a volume of culture with a volume of 1M NaOH, boiling for 5 min and removing cell debris by centrifugation at 7740× *g*.

### 4.6. Cytosolic Copper Quantification in Dormant Spores and Mycelium

Spores were obtained from SFM cultures. Young vegetative mycelium was obtained from sucrose-free R5A young cultures reaching a cellular mass between 0.009 and 0.012 mg protein per mL (16 or 20 h culture depending on the specific strain). Cytosolic copper quantification was performed as reported previously [13]. Briefly, the spores or the mycelium were washed four times by centrifugation at 12,000× *g* for 10 min at 4 °C and resuspended in a washing buffer (10 mM Tris-HCl pH 7.5 containing 1 mM EDTA). The samples were centrifuged and washed in the washing buffer, but this time without EDTA. For the bulk analysis of Cu in the dormant spores, acid digestion was conducted by resuspending the spores in 65% sub-boiling purified HNO_3_ at 70 °C for 1 h and then 30% H_2_O_2_ for 3 h at the same temperature. For the bulk analysis of Cu in the germinated spores and the mycelium, the samples were suspended in a rupture buffer (10 mM Tris-HCl pH 7.5) The lysis step was performed using Fast-Prep (MP™ Biomedicals Germany GmbH, Berlin, Germany) with six 20-s force 6.5 cycles and with 1 min on ice between each run. The cell debris was eliminated by centrifuging samples at 12,000× *g* for 10 min at 4 °C and discarding the pellets. The resulting solutions were diluted with water and the total Cu content was determined by ICP-MS and referred to the dry mass of the spores (1 mL of spores were washed with water, dried at 100 °C to a constant weight on pre-weighted tubes) or protein (measured with the Bradford assay).

All the measurements were conducted in the triple quadrupole-based ICP-MS Thermo iCAP-TQ (Thermo Fisher Scientific, Bremen, Germany) using the single quad mode and helium as collision gas. For the bulk analysis, the ICP was equipped with a Micro Mist nebuliser, a cyclonic spray chamber (both from ESI Elemental Service & Instruments GmbH, Mainz, Germany) and an auto-sampler ASX-560 (Teledyne CETAC Technologies, Omaha, NE, USA).

All the solutions were prepared using ultrapure water obtained from a Milli-Q system (Millipore, Bedford, MA, USA). The hydrogen peroxide for the acid digestion was obtained from Sigma-Aldrich (Saint Louis, MO, USA). The nitric acid (65%, suprapur quality) was purchased from Merck Millipore (Darmstad, Germany) and further purified by sub-boiling distillation. The external calibrations were carried out with a Cu ICP standard CertiPur^®^ (1000 mgL^−1^), purchased from Merck.

The reliability of the differences between the copper concentrations (compared to the wild-type strain) was analysed by Student’s *t*-tests. The differences were considered significant if the p value was equal to or less than 0.05 (asterisks in Figure 2c).

### 4.7. Quantification of Spore Germination

The *S. coelicolor* germination was quantified as previously described [13] in GYM [24] cellophane cultures inoculated with 10^7^ spores. Cellophane squares were manually cut, placed over a coverslip, stained with SYTO9 and propidium iodide (LIVE/DEAD Bac- Light Bacterial Viability Kit, Invitrogen, L1-3152) and observed under a Leica TCS-SP8 confocal laser-scanning microscope (Leica Microsistemas S.L.U., Spain) at wavelengths of 488 and 568 nm excitation and 530 nm (green) or 640 nm (red) emissions [30]. Three biological replicates were processed quantifying germination in at least 100 spores per replicate. Germination values were normalised as the percentage of germination reduction compared to the wild-type strain. The reliability of the differences in spore germination (compared to the wild-type strain) was analysed by Student’s *t*-tests. The differences were considered significant if the p value was equal to or less than 0.05 (asterisks in Figure 2).

### 4.8. Compound Extraction and HPLC-MS Metabolome Analyses

Three biological replicates were cultured and collected at the maximum secondary metabolite production time-points (145 h in the *S. coelicolor* strains; 48 h in the *S. venezuelae* strains). The cells and supernatants from the 20-mL cultures were separated by centrifugation at 10,000 rpm for 10 min. In the case of the *S. coelicolor*, supernatant cultures and cells were processed, whereas in the *S. venezuelae*, only the supernatant cultures were processed. The supernatant culture compound extraction was carried out by adding 0.56 volumes of ethyl acetate, vortexing three times for 2 min and centrifuging for 5 min at 10,000 rpm (the pellet was discarded). The cell extraction was carried out by adding one volume of acetone, followed by sonication in an ultrasonic bath (2 min on, 2 min off; three cycles), and centrifugation at 10,000 rpm for 10 min. The pellets were subjected to a second extraction using ethyl acetate (instead of acetone), following the same vortexing and centrifugation protocol, and the ethyl acetate supernatants were collected. The culture supernatant extracts (ethyl acetate) were vacuum-dried using rotary evaporation (RV 10 Digital, IKA®-Werke GmbH & Co. KG, Germany). The cell extracts (acetone and ethyl acetate) were added to the same evaporating flasks in which the supernatant culture extracts were dried, and vacuum-dried by rotary evaporation. The dry extracts (from the supernatant cultures and the cells in the *S. coelicolor*, and only the supernatant cultures in the *S. venezuelae*) were kept at −20 °C in the evaporating flasks.

The dry extracts were dissolved in 4 mL of methanol, transferred to 2-mL centrifugation tubes and dried in the speedvac. The pellets were dissolved and combined in 400 µL of methanol:DMSO (1:1, *v*/*v*). Five µL from each sample were analysed by liquid chromatography-electrospray ionisation mass spectrometry (LC-HRESI-MS). A UPLC system (Dionex Ultimate 3000, Thermo Scientific, Waltham, MA, USA), coupled with an ESI-UHR-Qq-TOF Impact II spectrometer (Bruker, Billerica, MA, USA), was used, with negative ion mode acquisition in a *m*/*z* range from 40 to 2000 Da. The chromatographic separation was carried out using a Zorbax^®^ Eclipse Plus C18 column (50 × 2.1 mm, 1.8 μm) (Agilent Technologies, Santa Clara, CA, USA).

The *S. coelicolor* analytes were eluted at a flow rate of 0.25 mL/min using a gradient of 0.1% (*v*/*v*) formic acid in water (mobile phase A) and 0.1% (*v*/*v*) formic acid in acetonitrile (mobile phase B), as follows: 0–10% (*v*/*v*) of B for 1 min, which was increased to 35% over 3 min, maintained at 35% for 1 min, then increased to 100% over 3 min, maintained at 100% during 2 min, decreased to 10% over 1 min and maintained at 10% (*v*/*v*) for 4 min.

The *S. venezuelae* analytes were eluted using the same flow rate and mobile phases as in the *S. coelicolor*, but the gradient was different: 0–5% (*v*/*v*) of B for 5 min, which was increased to 55% (*v*/*v*) over 25 min, and finally increased to 100% over 13 min.

### 4.9. S. albus HPLC-DAD Metabolome Analyses

The *S. albus* samples were collected from 108 h cultures. The compounds were extracted as indicated above for the *S. coelicolor*. The samples were injected into the Agilent 1260 Infinity instrument equipped with an analytical C-18 HPLC column (250 × 4.0 mm, Pursuit^®^ XRs 5, 5 μm, Agilent Technologies, Santa Clara, CA, USA) and coupled to a UV detector set to 190-640 nm at a flow rate of 1 mL/min over a period of 51 min, using H_2_O (A) and MeCN (B) as the mobile phase, both with 0.1% (*v*/*v*) of formic acid: 0–5 min (10% B), 5–22 min (10–35% B), 22–27 min (35% B), 27–44 min (35–100% B), 44–45 min (100% B), 45–46 min (100–10% B) and 46–51 min (10% B).

### 4.10. Bioinformatic Analysis of the Metabolome

The raw data were processed with the software package DataAnalysis (version 4.3, Bruker, Billerica, MA, USA ). The chromatograms were calibrated using internal calibrants (sodium, phosphate acetate). Both the culture media used for the fermentation contained MOPS, which creates an intense peak that is easily recognisable at the beginning of the chromatogram. The MOPS was used to test the reliability of the calibration, which had an average deviation of ±0.0002 (i.e., 0.2 ppm).

Chromatograms were processed using MZmine 2 software (G0 Cell Unit, Okinawa Institute of Science and Technology, Onna, Okinawa, Japan) [31]. Each sample was compared against the wild-type strain (three biological replicates each) according to the following workflow: mass detection was performed with the centroid mass detector using a noise level of 2.0E2; the ADAP chromatogram builder was set at a minimum group of 10, a group intensity threshold of 2.0E2, a minimum highest intensity of 5.0E2 and a *m*/*z* tolerance of 0.001; smoothing was performed with a filter width of 9; the chromatogram deconvolution used a local minimum search, a chromatogram threshold of 90%, a minimum RT range of 0.05, a minimum relative height of 1%, a minimum absolute height of 5.0E2, a minimum ratio of peak top/edge 2 and a peak duration range between 0.03 and 3, *m*/*z* center calculation median; the isotopes were grouped using a *m*/*z* tolerance of 0.001 and an RT tolerance of 0.1, a monoisotopic shape, a maximum charge of 2; the representative isotope was selected as the most intense; using a join aligner with a *m*/*z* tolerance of 0.001, weight for *m*/*z* of 20, RT tolerance of 0.1, weight for RT of 20, we compared the isotope pattern with a *m*/*z* tolerance of 0.001, minimum absolute intensity of 5.0E2 and a minimum score of 50%; and the duplicate peak filter used the single feature filter mode, an *m*/*z* tolerance of 0.001 and an RT tolerance of 0.1.

The MZmine results were exported to MetaboAnalyst 5.0 (Institute of Parasitology, McGill University, Montreal, QC, Canada) [32] for statistical analysis. To focus on the most reliable data, abundance values lower than 1e4 were discarded before statistical analysis. The MetaboAnalyst parameters were as follows: the missing values were estimated as 1/5 of the minimum positive value of each variable; the abundance values were log10-transformed; and the data were scaled by “pareto scaling”. The differences in abundance values (compared to the wild-type strain) were considered significant if the q-value was equal to or less than 0.05.

The secondary metabolites produced by the *Streptomyces coelicolor* M145 and the *Streptomyces venezuelae* NRRL B-65442 were deduced from Nett et al. [22] and Lee et al. [7] respectively. The formula and monoisotopic exact masses of these secondary metabolites were obtained from Pubchem (https://pubchem.ncbi.nlm.nih.gov/) (accessed on 30 June 2021). The exact masses of the ions of these compounds and their putative adducts were obtained according to Huang et al. [33]. This list of secondary metabolite masses was used to identify the putative secondary metabolites in the *S. coelicolor* and *S. venezuelae* metabolomes, using a mass tolerance threshold of ±0.001 Da. The retention time tolerance used to consider a specific mass as an isomer was 0.1 min (i.e., differences higher than 0.1 indicated different isomers).

## 5. Conclusions

In this study, we show that increasing the cytosolic copper concentration by means of the alteration of the *SCO2730/31* orthologue expression helps to modulate secondary metabolism in *S. coelicolor*, *S. venezuelae* and *S. albus*. The antisense *SCO2730/31* consensus mRNA created here constitutes a tool for the knockdown of the expression of the *SCO2730/31* genes which can contribute to the modification of secondary metabolism in streptomycetes and the activation of cryptic pathways, albeit to a lower level than in the *S. coelicolor SCO2730::Tn5062* mutant. Further work will be necessary to fully understand the complex regulation of the *SCO2730/31*’s gene expression and to reproduce the dose of the SCO2730/31 copper chaperone/transporter of the *S. coelicolor SCO2730::Tn5062* mutant in other streptomycetes.

## Figures and Tables

**Figure 1 ijms-22-10143-f001:**
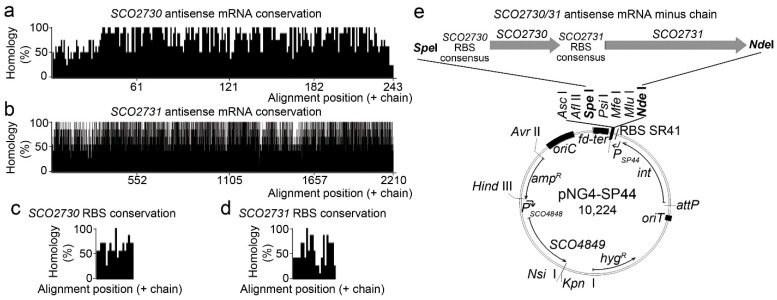
*SCO2730/31* consensus antisense mRNA conservation and overexpression. (**a**–**d**) Histogram showing the percentages of homology of the *SCO2730* and *SCO2731* ORFs and the *SCO2730* and *SCO2731* RBSs. (**e**) Diagram showing the pNG4-SP44 overexpression plasmid controlling the expression of the *SCO2730/31* antisense mRNA.

**Figure 2 ijms-22-10143-f002:**
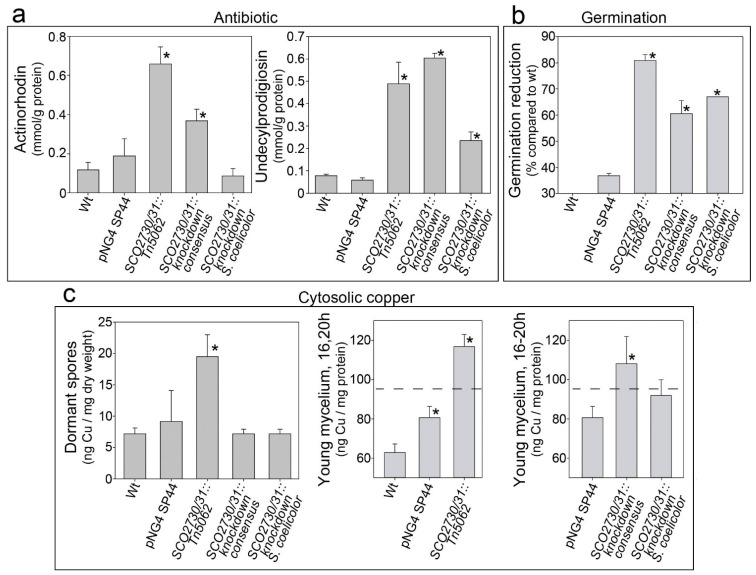
Antibiotic production, germination and cytosolic copper concentration in the *S. coelicolor SCO2730/31* knockdown mutants compared to the wild-type strain with and without the pNG4-SP44 plasmid and the *SCO2730::Tn5062* mutant. (**a**) Maximum actinorhodin and undecylprodigiosin production (145 h sucrose-free R5A liquid cultures). (**b**) Percentage of germination reduction compared to the wild-type strain (8 h GYM solid cultures). (**c**) Cytosolic copper in dormant spores (left panel, SFM cultures) and young mycelium (16–20 h sucrose-free R5A liquid cultures) (middle and right panels). The dashed line indicates the cytosolic copper umbral triggering secondary metabolism [13]. The asterisks indicate significant differences compared to the wild-type strain or the wild-type strain harbouring the pNG4-SP44 plasmid (*p* value equal to or less than 0.05).

**Figure 3 ijms-22-10143-f003:**
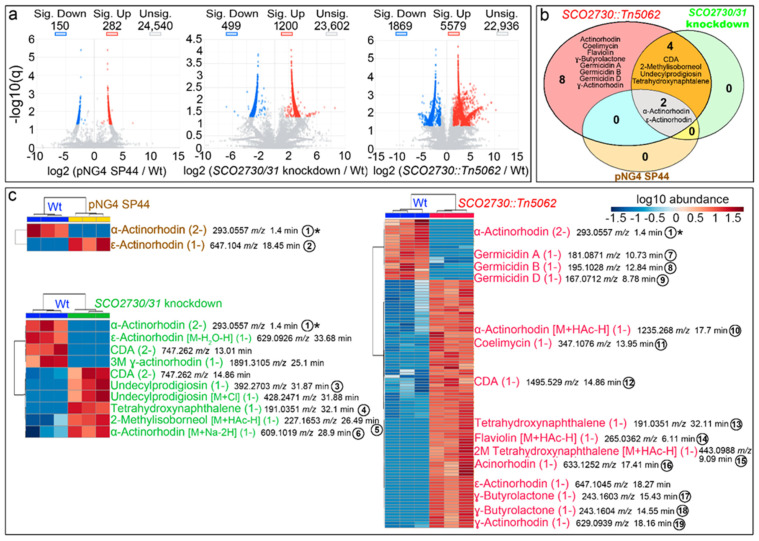
Metabolome analysis of the *S. coelicolor SCO2730::Tn5062* and the *SCO2730/31* knockdown mutants compared to the *S. coelicolor* wild-type strain with and without pNG4-SP44. (**a**) Volcano plots of the abundance values of the compounds detected in each strain compared to the wild-type strain. Significant up- (red) and down-regulated (blue) compounds are indicated. (**b**) Venn diagram showing the putatively identified secondary metabolites, showing significantly affected production in the *SCO2730::Tn5062* mutant, the *SCO2730/31* knockdown mutant and the pNG4-SP44 strains compared to the wild-type strain. (**c**) Heat-map showing the abundance values of putatively identified secondary metabolite adducts and isomers showing significant differences compared to the wild-type strain. The abundance values from three biological replicates are shown. The numbers indicate compounds detected in only one of the samples, whose chromatograms are indicated in Figure 4. The asterisk indicates the only compound identified in all the strains that was significantly down-regulated compared to the wild-type strain.

**Figure 4 ijms-22-10143-f004:**
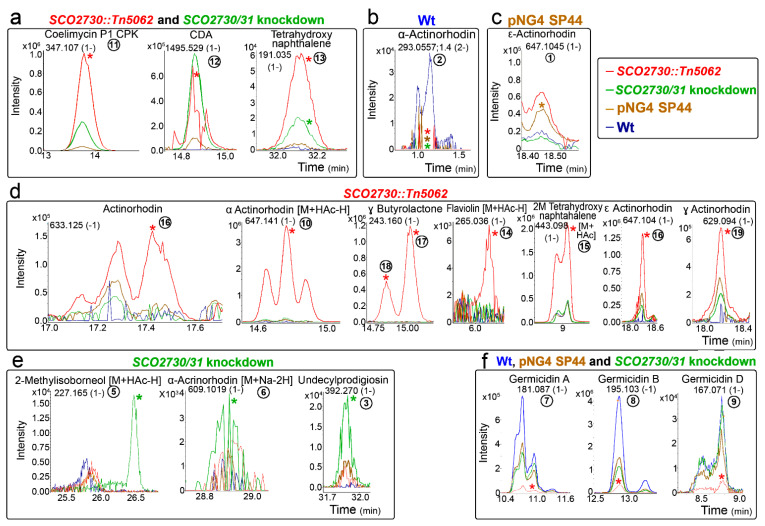
HPLC-MS chromatograms of some of the secondary metabolite compounds, showing different abundance in the *S. coelicolor SCO2730::Tn5062* and the *SCO2730/31* knockdown mutants compared to the *S. coelicolor* wild-type strain with and without pNG4-SP44. (**a**) Compounds showing higher abundances in the *SCO2730::Tn5062* and the *SCO2730/31* knockdown mutants. (**b**) Compound showing higher abundance in the *S. coelicolor* wild-type strain. (**c**) Compound showing higher abundance in the *S. coelicolor* wild-type strain harbouring pNG4-SP44. (**d**) Compounds showing higher abundances in the *SCO2730::Tn5062* mutant. (**e**) Compounds showing higher abundances in the *SCO2730/31* knockdown mutant. (**f**) Compounds showing higher abundances in the *S. coelicolor* wild-type strain with and without pNG4-SP44, and the *SCO2730/31* knockdown mutant. Only one biological replicate is shown. The asterisks indicate significant differences (compared to the wild-type strain). The numbers are as in Figure 3.

**Figure 5 ijms-22-10143-f005:**
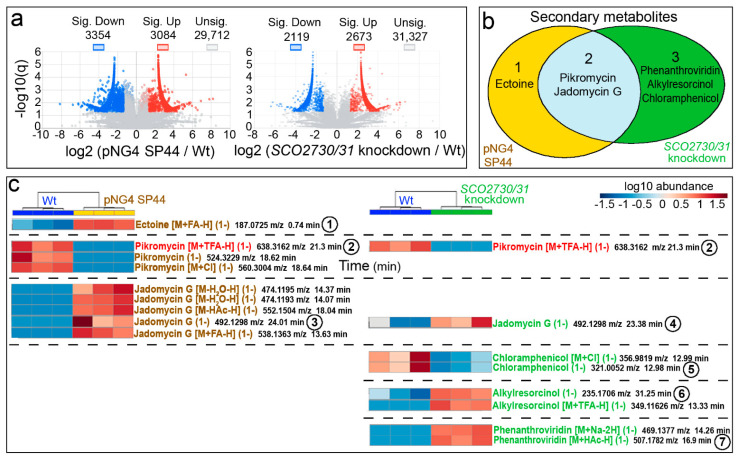
Metabolome analysis of the *S. venezuelae SCO2730/31* knockdown mutant compared to the *S. venezuelae* wild-type-strain SP44. (**a**) Volcano plots of the abundance of the compounds detected in each strain compared to the wild-type strain. Significant up- (red) and down-regulated (blue) compounds are indicated. (**b**) Venn diagram showing the putatively identified secondary metabolites, showing significantly affected production in the *SCO2730/31* knockdown mutant and the pNG4-SP44 strains compared to the wild-type strain. (**c**) Heat map showing the abundance values of the putatively identified secondary metabolite adducts and isomers, showing significant differences compared to the wild-type strain. The abundance values from three biological replicates are shown. The numbers indicate compounds whose chromatograms are indicated in Figure 6. The only adduct identified in the pNG4-SP44 and the SCO2730/31 strains is highlighted in red.

**Figure 6 ijms-22-10143-f006:**
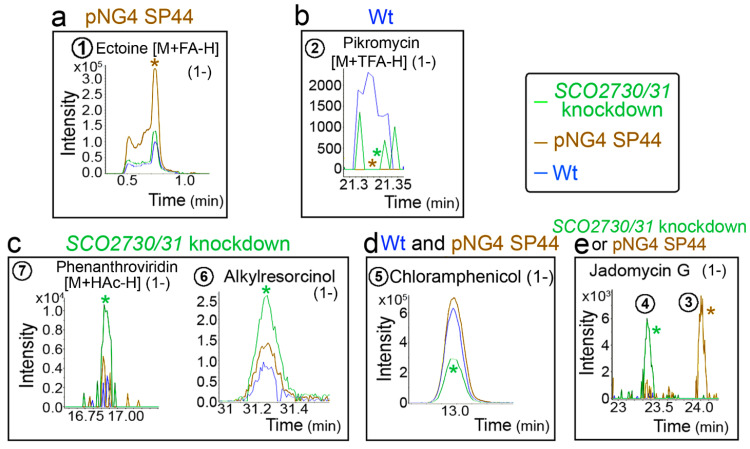
HPLC-MS chromatograms of some of the identified secondary metabolite compounds, showing different abundance values in the *S. venezuelae SCO2730/31* knockdown mutant compared to the *S. venezuelae* wild-type strain with and without pNG4-SP44. (**a**) Compound showing higher abundance in the *S. venezuelae* strain harbouring pNG4-SP44. (**b**) Compound showing higher abundance in the *S. venezuelae* wild-type strain. (**c**) Compounds showing higher abundances in the *S. venezuelae SCO2730/31* knockdown mutant. (**d**) Compounds showing higher abundances in the *S. venezuelae* wild-type strain with and without pNG4-SP44. (**e**) Jadomycin G isomers detected in the *S. venezuelae* strain harbouring pNG4-SP44 and the *SCO2730/31* knockdown mutant. Only one biological replicate is shown. The asterisks indicate significant differences. The numbers are as in Figure 5.

**Figure 7 ijms-22-10143-f007:**
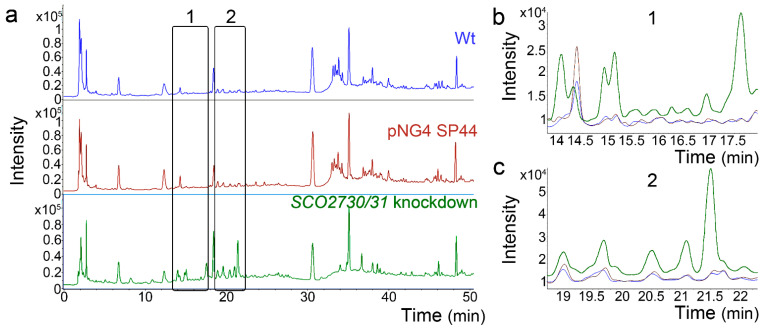
HPLC-DAD chromatograms of the *S. albus SCO2730/31* knockdown mutant compared to the *S. albus* wild-type strain with and without pNG4-SP44. (**a**) Whole chromatograms. The rectangles mark the clearest differences between the *SCO2730/31* knockdown mutant and the wild-type strain with or without pNG4-SP44. (**b**,**c**) Magnifications of the chromatogram areas marked with rectangles. Only one biological replicate is shown.

## Data Availability

The LC-MS raw data are available upon request.

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
