# Peer review of "The Modulation of SCO2730/31 Copper Chaperone/Transporter Orthologue Expression Enhances Secondary Metabolism in Streptomycetes"

_ijms, 2021, doi:10.3390/ijms221810143_

Round 1
Reviewer 1 Report
In this work, the authors describe the modulation of SCO2730/31 copper chaperone/transporter orthologue expression enhances secondary metabolism in streptomycetes. I very much enjoyed this manuscript and am very happy to have been able to read it
Here my consideration:
- I suggest changing the colors of the figures by choosing darker ones and avoiding yellow, because they make the figures very difficult to read
- move table 1 to supplemental files
- separate the conclusions paragraph from the discussion, thus making two separate paragraphs
Author Response
RESPONSES TO THE REVIEWER 1
Note: All modifications were marked up using the MS Word “Track Changes” function. In order to facilitate revision, modifications made following the reviewer’s comments were highlighted in yellow.
REVIEWER 1
“move table 1 to supplemental files”
“separate the conclusions paragraph from the discussion, thus making two separate paragraphs”
Response 1. As suggested, darker colours were used in the revised figures, yellow was replaced by brown, table 1 was moved to supplemental files, conclusions were separated from discussion.

Reviewer 2 Report
This study was performed to reproduce the effect of knockout of the copper export genes of Streptomyces coelicolor SCO2730 (copper chaperone, SCO2730::Tn5062) by using antisense mRNA based gene knockdown approch.
Authors applied antisense mRNA to knockdown homologous gene in S. venezuelae and studied the impact of gene knockdown on secondary metabolite production and modulation of coper homeostasis.
This study is important to discover new antibiotics but need improvement. Below are the general comments which may help to improve the study.
- To prove the efficient target knockdown, authors need to show the protein levels of the target genes.
- Transcriptome profiles of knockout and knockdown mutants are not closely comparable. Maybe need to improve the knockdown system.
- Impact of knockdown in venezuelae is not closely comparable with Streptomyces coelicolor SCO2730 (copper chaperone, SCO2730::Tn5062).
- Authors need to show the impact of knockdown in multiple species to show that knockdown is a reliable tool and it work in multiple strains of Streptomyces.
Author Response
RESPONSES TO THE REVIEWER 2
Note: All modifications were marked up using the MS Word “Track Changes” function. In order to facilitate revision, modifications made following the reviewer’s comments were highlighted in yellow.
REVIEWER 2
“1. To prove the efficient target knockdown, authors need to show the protein levels of the target genes”
Response 1. To quantify the SCO2730 and SCO2731 protein levels would be indeed interesting. Unfortunately, we could not make a direct quantification of the SCO2730/31 protein amounts, because we have not antibodies against these proteins, and other alternative methods, as quantitative proteomics (for instance, selected reaction monitoring, SRM), would require very specialised personnel and equipment, as well as to optimise complex protocols. However, we quantified significant differences in cytosolic copper levels between the mutants and the wild-type strain (Figure 2c), which is an indirect indicator of the differences in the SCO2730/31 copper transporter activity between the mutants and the wild-type strain.
This point has been now stressed in the discussion (highlighted in yellow).
“2. Transcriptome profiles of knockout and knockdown mutants are not closely comparable. Maybe need to improve the knockdown system”
Response 2. We guess that the reviewer refers to the metabolome profiles, that are indeed different in the SCO2730::Tn5062 insertional mutant and the S. coelicolor knockdown strains. The SCO2730/31 consensus antisense mRNA is useful to alter secondary metabolism, albeit to a lower level than in the S. coelicolor SCO2730::Tn5062 mutant. The reviewer is right about the fact that we need to redesign our strategy to fully reproduce the dose of the SCO2730/31 proteins present in the SCO2730::Tn5062 mutant in industrial streptomycetes. For that, we will need to further understand the complex regulation of the SCO2730/31 expression, which is modulated by at least four promoters (González-Quiñónez et al., Sci Rep. 2019 9:4214). Moreover, the SCO2731 gene is essential since its expression can be reduced, but not completely eliminated (our recent unpublished results). Further unknown regulatory networks cannot be discarded, since different combinations of the four promoters together with the SCO2102/03 ORFs, could not fully restore the SCO2730/31 transcription in the SCO2730::Tn5062 mutant.
These important points are now stressed in the revised manuscript (paragraphs highlighted in yellow in the discussion and the conclusions).
“3. Impact of knockdown in venezuelae is not closely comparable with Streptomyces coelicolor SCO2730 (copper chaperone, SCO2730::Tn5062).
- Authors need to show the impact of knockdown in multiple species to show that knockdown is a reliable tool and it work in multiple strains of Streptomyces”
Response 3. The effect of the antisense mRNA in S. venezuelae is certainly lesser than in S. coelicolor. Recently, we used our SCO2730/31 consensus antisense mRNA in Streptomyces albus, observing an important effect in the metabolome. The S. albus results were included and discussed in the revised manuscript (Figure 7, 2.4 paragraph and last paragraph of the discussion).

Round 2
Reviewer 2 Report
Authors improved the revised manuscript by including metabolomics data and additional discussion.